# Effects of SGLT2 Inhibitors on Atherosclerosis: Lessons from Cardiovascular Clinical Outcomes in Type 2 Diabetic Patients and Basic Researches

**DOI:** 10.3390/jcm11010137

**Published:** 2021-12-27

**Authors:** Jing Xu, Taro Hirai, Daisuke Koya, Munehiro Kitada

**Affiliations:** 1Department of Diabetology and Endocrinology, Kanazawa Medical University, Uchinada, Ishikawa 920-0293, Japan; xujing@kanazawa-med.ac.jp (J.X.); taro-h@kanazawa-med.ac.jp (T.H.); koya0516@kanazawa-med.ac.jp (D.K.); 2Division of Anticipatory Molecular Food Science and Technology, Medical Research Institute, Kanazawa Medical University, Uchinada, Ishikawa 920-0293, Japan; 3Omi Medical Center, Omi General Hospital, Kusatsu, Shiga 525-8585, Japan

**Keywords:** SGLT2 inhibitors, atherosclerosis, cardiovascular disease, diabetes

## Abstract

Atherosclerosis-caused cardiovascular diseases (CVD) are the leading cause of mortality in type 2 diabetes mellitus (T2DM). Sodium-glucose cotransporter 2 (SGLT2) inhibitors are effective oral drugs for the treatment of T2DM patients. Multiple pre-clinical and clinical studies have indicated that SGLT2 inhibitors not only reduce blood glucose but also confer benefits with regard to body weight, insulin resistance, lipid profiles and blood pressure. Recently, some cardiovascular outcome trials have demonstrated the safety and cardiovascular benefits of SGLT2 inhibitors beyond glycemic control. The SGLT2 inhibitors empagliflozin, canagliflozin, dapagliflozin and ertugliflozin reduce the rates of major adverse cardiovascular events and of hospitalization for heart failure in T2DM patients regardless of CVD. The potential mechanisms of SGLT2 inhibitors on cardioprotection may be involved in improving the function of vascular endothelial cells, suppressing oxidative stress, inhibiting inflammation and regulating autophagy, which further protect from the progression of atherosclerosis. Here, we summarized the pre-clinical and clinical evidence of SGLT2 inhibitors on cardioprotection and discussed the potential molecular mechanisms of SGLT2 inhibitors in preventing the pathogenesis of atherosclerosis and CVD.

## 1. Introduction

Atherosclerosis is a chronic progressive disease characterized by the accumulation and deposition of lipids and fibrous elements in large arteries [1]. Generally, the formation of atherosclerotic plaques is divided into four stages: fatty streaks, atheromatous plaques, complicated atheromatous plaques and clinical complications [2]. Plaque rupture and thromboembolism may lead to severe cardiovascular diseases (CVD), including acute coronary syndrome (ACS), myocardial infarction (MI) or stroke [1,3]. CVD caused by atherosclerosis is the main cause of mortality in metabolic-related diseases, especially in type 2 diabetes mellitus (T2DM) [4,5,6]. Correspondingly, compared with non-diabetic individuals, T2DM patients have a higher risk of atherosclerosis and CVD [7,8]. Over the last decade, CVD globally affected approximately 32.2% of T2DM patients, accounting for 9.9% of deaths among them [9]. Therefore, in recent years, clinical trials of anti-diabetic agents have not only focused on hypoglycemic benefits but on cardiovascular safety and cardioprotective effects as well.

Sodium-glucose cotransporter 2 (SGLT2) is mainly localized in the proximal tubule, which is responsible for reabsorbing 80–90% of filtered glucose under physiological conditions [10,11]. The direct effect of SGLT2 inhibitors is to block glucose transportation in the kidney, which may depend on two distinct mechanisms: (1) by augmenting renal glucose excretion and (2) by ameliorating glucotoxicity [10]. Multiple clinical trials have demonstrated that SGLT2 inhibitors reduce blood glucose by inhibiting the reabsorption of filtered glucose in the proximal tubules independent of insulin. SGLT2 inhibitors slow the progression of macroalbuminuria, reduce the doubling of the serum creatinine level, sustain 40% reduction in the estimated glomerular filtration rate (eGFR) and further protect from inevitable renal replacement therapy or death from renal disease [12,13,14,15,16]. These renoprotective benefits of SGLT2 inhibitors may be attributed to glycemic control, reducing body weight, lipid profiles, blood pressure and uric acid and improving insulin resistance. Moreover, due to the improvement of these cardiovascular risk factors, SGLT2 inhibitors also present cardioprotective effects [10,11,12,13,14,15,16,17,18,19].

In this review, we summarized the clinical evidence of SGLT2 inhibitors on cardioprotection and discussed the potential molecular mechanisms of SGLT2 inhibitors in preventing the pathogenesis of atherosclerosis and CVD.

## 2. Pathophysiological and Pharmacologic Roles of SGLT2 Inhibition

The kidney is a crucial organ for maintaining glucose metabolism. In healthy individuals, the human kidney filters more than 180 g of glucose per day. SGLT2 is a 75 kilodalton (kDa) protein coded by the solute carrier family 5 (SLC5A2) gene in humans and is mainly localized in the kidney proximal tubule [20]. A total of 80–90% of filtered glucose is reabsorbed by SGLT2 in the early proximal tubule, while the remaining 10–20% is absorbed by SGLT1. The energy for the transport capacity of SGLT2 and SGLT1 is derived from the Na^+^/K^+^ ATPase pump located in the basolateral membrane of the proximal tubule. Therefore, the reabsorption of filtered glucose is also accompanied by Na^+^ reabsorption [10,18,21]. SGLT2 knockout mice had glucosuria and polyuria compared with wildtype mice [22]. In patients with T2DM, the reabsorption of filtered glucose in the proximal tubules is significantly increased, contributing to elevated SGLT2 expression [23,24]. Meanwhile, the elevated reabsorption of Na^+^ activates the local renin-angiotensin system, stimulating the constriction of the adjacent efferent arteriole and dilation of the afferent arteriole, which leads to increases in intraglomerular pressure and glomerular filtration rate (GFR), which ultimately results in glomerular damage [10]. This evidence indicates that the inhibition of SGLT2 is a potential target for glycemic control and conferring renoprotection in treating T2DM.

The first natural SGLT inhibitor, called phlorizin, is derived from fruit trees. Phlorizin functions as a competitive inhibitor of SGLT1/2 to block glucose absorption in the proximal renal tubule and mucosa of the small intestine [25,26]. However, due to poor solubility in water, poor oral bioavailability, the non-selective inhibition of both SGLT1 and SGLT2 and a short half-life, effective studies on the mechanisms of specific SGLT2 inhibition by phlorizin are limited [25,27]. Subsequently, more novel selective SGLT2 inhibitors have been identified. Compared with phlorizin, these SGLT2 inhibitors are structurally improved. Dapagliflozin contains C-glucoside, while canagliflozin and empagliflozin contain C-glycosylated diarylmethane pharmacophore. These structures are resistant to hydrolysis by β-glucosidases to increase their half-life and specificity to SGLT2 [27,28]. To date, empagliflozin, canagliflozin, dapagliflozin and ertugliflozin have been approved by the United States Food and Drug Administration (FDA) and the European Medicines Agency (EMA) for the treatment of T2DM [26]. Some other novel SGLT2 inhibitors, such as ipragliflozin and luseogliflozin, have been approved in Japan and other countries [29,30].

## 3. Mechanisms of SGLT2 Inhibitors against Atherosclerosis

During the formation of atherosclerotic plaques, three types of cells, endothelial cells (ECs), vascular smooth muscle cells (VSMCs) and monocytes/macrophages, play crucial roles. At the first stage, due to endothelial dysfunction, excessive lipids and lipoproteins accumulate in the subendothelial matrix. This process is triggered by the accumulation of oxidized low-density lipoprotein (Ox-LDL), which stimulates the release of pro-inflammatory cytokines, including interleukin 8 (IL-8) and adhesion molecules such as intercellular cell adhesion molecule-1 (ICAM-1), Vascular cell adhesion protein 1 (VCAM-1), monocyte chemoattractant protein 1 (MCP-1) and macrophage colony-stimulating factor (M-CSF). Then, circulating monocytes migrate to the intima, where they proliferate, differentiate into macrophages and combine with lipoproteins to form foam cells. Monocytes/macrophages express pro-inflammatory cytokines, including IL-1 and tumor necrosis factor α (TNFα). VSMCs migrating from the medial layer can also secret pro-fibrosis molecules. With the continuous migration of monocytes into the intima and the accumulation of lipids and lipoproteins, foam cells and SMCs die to form necrotic cores. Platelets will recruit to form thrombi and eventually develop atherosclerotic plaques. Vascular occlusion or plaque rupture may result in acute cardiovascular and cerebrovascular events [1,3,31].

Some metabolic characteristics, including obesity, dyslipidemia, hyperglycemia and hypertension, are independent risk factors for atherosclerosis [32]. These metabolic disorders, which may lead to endothelial dysfunction, oxidative stress, inflammation and the impairment of autophagy, participate in the formation of plaques and the pathogenesis of atherosclerosis [33]. A series of basic studies have confirmed that SGLT2 inhibitors have cardioprotective effects beyond glucose control in animal models of atherosclerosis [34,35,36,37,38]. The underlying mechanisms of SGLT2 inhibitors may be related to protecting endothelial function, anti-oxidative stress, anti-inflammation and maintaining basal and adaptive autophagy (Figure 1).

### 3.1. Maintaining Endothelial Function

Endogenous nitric oxide (NO) is a vasodilatory molecule produced by NO synthase (NOS), which presents antiatherosclerotic effects [39]. SGLT2 inhibitors inhibit cardiac Na^+^/K^+^ exchange to induce vasodilation [40]. Previous studies have shown that SGLT2 inhibitors can participate in the regulation of vasodilation by increasing the anabolism and bioavailability of NO. Both empagliflozin and canagliflozin ameliorated aortic stiffness and improved vasodilation via endothelial NOS (eNOS) phosphorylation in db/db mice and non-diabetic rats [41,42]. Empagliflozin and dapagliflozin restored NO bioavailability by suppressing reactive oxygen species (ROS) generation in TNFα-induced ECs [43].

### 3.2. Anti-Oxidative Stress 

An imbalance between the production and scavenging of ROS results in oxidative stress, which plays crucial roles in the progression of atherosclerosis [44]. Multiple risk factors of atherosclerosis, such as high glucose, elevated free fatty acids (FFA) and triglyceride (TG), can increase ROS production via activating nicotinamide adenine dinucleotide phosphate (NADPH) oxidases (NOX1-5, p22phox, p47phox), inhibiting NOS and suppressing glyceraldehyde 3-phosphate dehydrogenase (GAPDH) activity, and then by further amplifying oxidative stress in macrophages, ECs and adipocytes and accelerating the formation of atherosclerotic plaques [45,46,47,48]. 

Both in vitro and in vivo experiments have confirmed the anti-oxidative stress effects of SGLT2 inhibitors. Angiotensin II can induce SGLT2 expression in ECs, which leads to the activation of oxidative stress due to NADPH oxidase. Empagliflozin presented anti-oxidant effects to protect against endothelial senescence and dysfunction via suppressing the NADPH oxidase/SGLT2 pathway [49]. Canagliflozin reduced the expression of NADPH oxidase, including NOX2, NOX4, p22phox and p47phox, and reduced the urinary excretion of 8-hydroxy-2′ -deoxyguanosine (8-OHdG) in diabetic Apolipoprotein E-deficient (ApoE^−/−^) mice [50]. Both empagliflozin and dapagliflozin inhibited the production of ROS, ameliorated TNFα-induced oxidative stress and restored impaired NO bioavailability in human umbilical vein endothelial cells (HUVECs) and human coronary arterial endothelial cells (HCAECs) [43]. SGLT2 inhibitors, including canagliflozin, dapagliflozin and empagliflozin, protected from high-glucose-induced vasodilation disorders via suppressing NAPDH oxidase/ROS signaling in cultured ECs [51]. 

### 3.3. Anti-Inflammation

Inflammatory cytokines and inflammatory cascade reactions throughout participate in the formation, progression and rupture of atherosclerotic plaque [1,31]. Previous studies have revealed that SGLT2 inhibitors, including dapagliflozin, empagliflozin, canagliflozin and luseogliflozin, reduced a series of pro-inflammatory cytokines, including IL-1β, IL-18 [34], IL-6 and TNFα [35], and adhesion molecules such as MCP-1 [35,38,52,53], ICAM-1 [36,50] and VCAM-1 [50,52,53] in atherosclerosis animal models with or without diabetes. In vitro studies have explored the potential mechanisms of the anti-inflammatory effects of SGLT2 inhibitors. Dapagliflozin may inhibit high glucose-induced TNFα, MCP-1 and VCAM-1 via suppressing the nuclear factor kappa-B (NF-κB) pathway in human vascular endothelial cells [54]. Canagliflozin suppressed IL-1β-activated cytokine and chemokine, such as IL-6 and MCP-1, partly via AMP-activated protein kinase (AMPK) activation [55]. 

In addition to inflammatory pathways, inflammasome-mediated inflammatory pathways are also involved in the pathogenesis of atherosclerosis [56,57,58]. Ox-LDL and high glucose are responsible for the activation of the nucleotide-binding oligomerization domain-like receptor family pyrin domain containing 3 (NLRP3) inflammasome [56,59]. NLRP3 is essential for activating the precursors of IL-1β and IL-18 into their mature forms, which recruit in vascular endothelial cells, leading to atherothrombosis [58]. NF-κB promotes the transcription of NLRP3 and participates in cardiac inflammation [60], which links inflammasome and the inflammatory pathway. Moreover, TG and very low-density lipoprotein (VLDL)-related arterial inflammation are closely related to the nucleotide-binding oligomerization domain-like receptor family pyrin domain containing 1 (NLRP1) inflammasome activation in ECs [61]. Dapagliflozin inhibited IL-1β expression via NLRP3/caspase 1 signaling in streptozotocin (STZ)-induced diabetic ApoE^−/−^ mice and T2DM rodent models [34,62,63]. Empagliflozin suppressed IL-17A-induced IL-1β and IL-18 secretions via NLRP3/caspase1 signaling and further inhibited the cell proliferation and migration of human aortic SMCs [64] and decreased the expression of IL-1β via suppressing NF-κB phosphorylation/NLRP3 signaling in human macrophages [62].

### 3.4. Regulation of Autophagy

Autophagy plays a complex role in the pathogenesis of atherosclerosis. Basal and mild adaptive autophagy to stress can maintain the endothelial functions of ECs, VSMCs and macrophages and protect against the formation of atherosclerotic plaques, while both deficient and excessive autophagy are related to inflammation, oxidative stress and apoptosis, which may contribute to autophagy-dependent cell death, aggravate vascular injury and lead to plaque instability or rupture [33,65,66]. Therefore, the precise regulation of autophagy is essential to prevent the development of atherosclerosis.

The SGLT2 inhibitors empagliflozin and dapagliflozin have been identified to restore autophagy deficiency in diabetic or obese rodent models, which mainly depends on the activation of nutrient-sensing pathways, such as the AMPK/mTOR (mechanistic target of rapamycin) signaling pathway [67,68]. In vitro studies have also identified several potential mechanisms. Empagliflozin restored autophagic flux impairment via activating AMPK and suppressing mTOR in H9c2 cells (rat cardiac myoblast) [69], RAW264.7 and THP-1 cells (macrophage cell lines) [70]. Our previous study also indicated that dapagliflozin had similar effects on autophagy restoration via AMPK/mTOR signaling in high-glucose-treated proximal tubular cells [71].

## 4. Improvements of Indicators Related to CVD by SGLT2 Inhibitor

Disorders of metabolic-related characteristics, including glucose, lipid profiles, blood pressure, uric acid, etc., contribute to increased risks of developing atherosclerosis and CVD [33]. Beyond the benefits from glycemic control, multiple clinical and rodent research studies have demonstrated that the cardiovascular benefits of SGLT2 inhibitors are closely related to the improvement of these metabolic-related characteristics.

### 4.1. Pre-Clinical Evidence

ApoE^−/−^ mice and low density lipoprotein receptor knockout (Ldlr^−/−^) mice are well-established and extensively used rodent models of atherosclerosis [72]. Multiple studies have demonstrated that SGLT2 inhibitors, including dapagliflozin, empagliflozin, canagliflozin, luseogliflzin and ipragliflozin, reduce metabolic indicators to protect from the progression of atherosclerosis (Table 1). Dapagliflozin reduced fasting blood glucose (FBG), total cholesterol (TC) and TG [34], body weight, and glycosylated hemoglobin (HbA1c) [73] in STZ-induced diabetic ApoE^−/−^ mice and STZ-induced diabetic Ldlr^−/−^ mice [37]. Empagliflozin ameliorated FBG, TC, heart rate, blood pressure (BP) [53], TG, LDL [74], urinary microalbumin, body weight and fat mass [35,74] in high fat diet (HFD)-fed ApoE^−/−^ mice and reduced body weight and TG in STZ-induced diabetic ApoE^−/−^ mice [38]. Canagliflozin decreased glucose, TC, TG, LDL and heart rate in HFD-fed diabetic ApoE^−/−^ mice [52] and reduced TC and glucose in STZ-induced diabetic ApoE^−/−^ mice [50]. Luseogliflozin reduced body weight, TC, TG, BP and glucose in HFD-fed mice [75] and nicotinamide in STZ-induced diabetic ApoE^−/−^ mice [36]. Ipragliflozin decreased body weight and glucose to inhibit vascular remodeling in HFD-fed mice [76]. All these metabolic changes ameliorate vascular remodeling, reduce plaque size and increase plaque stability to protect from the progression of atherosclerosis.

In addition to these risk factors of CVD, T2DM patients treated with SGLT2 inhibitors showed elevated plasma ketone levels [77]. SGLT2 inhibitors reduce plasma glucose concentration by increasing renal glucose excretion. To meet the energy requirements of cellules, lipid oxidation is activated, leading to the increasing of acetyl coenzyme A (CoA). Acetyl CoA can enter the Krebs cycle to be converted to ketones (acetoacetate and β-hydroxybutyrate). Moreover, enhanced lipolysis in the adipocyte leads to the increase of plasma FFA, which can also be converted to acetyl CoA via β oxidation and subsequently to ketones in the liver [10]. The elevated ketones may improve the energy metabolism of the heart. Previous studies indicated that β-hydroxybutyrateas is a strong anti-inflammatory factor. Empagliflozin can significantly increase serum β-hydroxybutyrateas to inhibit NLRP3 inflammasome and reduce the expression of IL-1β levels, which benefit from CVD [62] (Figure 2).

### 4.2. Clinical Evidence

SGLT2 inhibitors not only reduce high glucose independent of insulin but also improve systolic blood pressure (SBP), body weight, lipid profile and uric acid, which are considered high risk factors for CVD [10] (Figure 2). The improvement of these metabolic indicators may play a crucial role in protecting against the pathogenesis of atherosclerosis and CVD. Numerous clinical trials have confirmed that SGLT2 inhibitors can effectively reduce body weight, which may be related to increases in fat utilization, reductions in adipose tissue mass and browning in white adipose tissue, further attenuating obesity-induced insulin resistance [78,79,80,81,82]. Treatment with empagliflozin [83,84,85], canagliflozin [82,86,87,88] and ertugliflozin [89] reduced SBP in patients with T2DM and hypertension or patients with T2DM and chronic kidney disease (CKD). This benefit may be related to minimal natriuresis and urinary glucose excretion, which lead to weight loss, osmotic diuresis, reduced plasma volume and arterial stiffness [90]. SGLT2 inhibitors also lead to a small decrease in plasma TG, increases in high-density lipoprotein cholesterol (HDL-C) and a small increase in LDL [82,91,92,93]. A potential mechanism of increased LDL and decreased TG with SGLT2 inhibition may be related to the reduction of LDL clearance, the greater lipolysis of triglyceride-rich lipoproteins [94] and the effect of switching energy metabolism from carbohydrate to lipid utilization [95]. Elevated circulating uric acid is associated with the risk of hypertension and CVD. Multiple studies have demonstrated that SGLT2 inhibitors can reduce circulating uric acid via enhancing urinary uric acid excretion in association with increased urinary glucose [96,97,98], which may benefit patients with CVD.

## 5. Clinical Evidence of SGLT2 Inhibitors against CVD

Since 2008, the FDA has required that cardiovascular outcome trials (CVOTs) be carried out for new anti-diabetic drugs due to the significance of cardioprotection on T2DM patients. All potential drugs should exclude the major adverse cardiovascular events (MACE) defined in the FDA guidance, including non-fatal myocardial infarction, non-fatal stroke and cardiovascular death [99,100]. Furthermore, the effect of SGLT2 inhibitors on cardiovascular outcomes, including hospitalization for heart failure, death and CVD, compared to other glucose-lowering drugs, was also verified in a real clinical practice [101,102].

### 5.1. Randomized Controlled Trials (RCTs)

A meta-analysis that summarized 6 regulatory submissions and 57 published trials revealed that seven different SGLT2 inhibitors reduced the risk of Major Adverse Cardiovascular Events (MACEs) and death from any cause [103]. In this section, we summarized some landmark CVOTs of SGLT2 inhibitors (Table 2).

The Empagliflozin Cardiovascular Outcome Event Trial in Type 2 Diabetes Mellitus Patients–Removing Excess Glucose trial (EMPA-REG OUTCOME) was the first CVOT to determine the cardiovascular benefits of SGLT2 inhibitors. This trial enrolled T2DM patients, who were treated with either empagliflozin or placebo. The primary composite outcome was MACEs. Compared with the placebo group, the empagliflozin group showed a significant reduction in death from MACEs, hospitalization for heart failure and death from any cause. However, there were no significant benefits for MI and stroke [104]. Subsequently, two trials demonstrated cardiovascular benefits in patients with heart failure and a reduced ejection fraction. The Empagliflozin Outcome Trial in Patients with Chronic Heart Failure with Reduced Ejection Fraction (EMPEROR-Reduced) enrolled 3730 patients (50% diagnosed as diabetes) with New York Heart Association (NYHA) class II to IV heart failure or with an ejection fraction (EF) ≤40%. Compared with the placebo group, the empagliflozin group had a lower risk of any inpatient or outpatient worsening heart failure event regardless of T2DM [105,106]. The Empagliflozin Outcome Trial in Patients with Chronic Heart Failure with Preserved Ejection Fraction (EMPEROR-Preserved) trial also identified the benefits of empagliflozin on heart failure in patients with NYHA class II to IV and an EF ≥40% [107].

The cardiovascular benefits of canagliflozin were demonstrated by the Canagliflozin Cardiovascular Assessment Study (CANVAS) and Canagliflozin and Renal Events in Diabetes with Established Nephropathy Clinical Evaluation (CREDENCE). CANVAS involved patients with T2DM, who were treated with canagliflozin or placebo. The primary outcome was also MACEs. Similar to empagliflozin, canagliflozin showed a significant reduction in MACEs, hospitalization for heart failure and death from any cause [19]. CREDENCE confirmed the cardio and renal benefits of canagliflozin in patients with T2DM and chronic kidney disease (CKD). The primary outcome was a composite of end-stage kidney disease or death from renal or cardiovascular disease. Patients in the canagliflozin group had a lower risk of cardiovascular death, MI, stroke and hospitalization for heart failure [14].

The Dapagliflozin Effect on Cardiovascular Events–Thrombolysis in Myocardial Infarction 58 (DECLARE–TIMI 58) evaluated the cardiovascular benefits of dapagliflozin on T2DM patients. The primary outcome was MACEs. Although dapagliflozin was not found to reduce the risk of MACEs, it reduced the risk of hospitalization for heart failure [108]. Another trial, named the Dapagliflozin and Prevention of Adverse Outcomes in Heart Failure (DAPA-HF) trial, confirmed this benefit. DAPA-HF enrolled patients with heart failure and reduced ejection fraction. The primary outcome was a composite of worsening heart failure or death from cardiovascular causes. This trial also demonstrated that dapagliflozin reduced the risk of heart failure hospitalization and cardiovascular death regardless of diabetes [109]. 

Ertugliflozin has also completed the assessment of cardiovascular events recently. The Evaluation of Ertugliflozin Efficacy and Safety Cardiovascular Outcomes Trial (VERTIS CV) enrolled patients with T2DM and established CVD. The primary outcome was MACEs. Similar to dapagliflozin, ertugliflozin was not found to reduce the risk of MACEs, but it did reduce the risk of hospitalization for heart failure [110].

In addition to the above SGLT2 inhibitors approved by the FDA and EMA for cardiovascular indications, another SGLT2 inhibitor, sotaliflozin, also present benefits for cardiovascular death and heart failure. The Effect of Sotagliflozin on Cardiovascular Events in Patients with Type 2 Diabetes Post Worsening Heart Failure (SOLOIST-WHF) and the Effect of Sotagliflozin on Cardiovascular and Renal Events in Patients with Type 2 Diabetes and Moderate Renal Impairment Who Are at Cardiovascular Risk (SCORED) trial demonstrated that, compared with placebo, sotaliflozin treatment significantly reduced the total number of deaths from cardiovascular causes and hospitalizations and urgent visits for heart failure in T2DM patients with recent worsening heart failure [111] and T2DM patients with chronic kidney disease regardless of albuminuria [112].

According to these hallmark CVOTs, the effects of SGLT2 inhibitors on atherosclerotic cardiovascular events, such as MI and strokes, are less impressive than the effects on heart failure (Table 2). A possible reason is the heterogeneity in the CVD risk of the study populations. Besides, SGLT2 inhibitors may primarily target on ameliorating cardiac structure and function (“the pump”) and not on the “pipes” (coronary arteries) [113,114]. Patients with T2DM and cardiac hypertrophy, diastolic or systolic dysfunction or with a hypervolemic state (regardless of cardiac or renal origin) are the best candidates for treatment with SGLT2 inhibitors [114].

### 5.2. Multinational Observational Cohort Study

The effects of drugs may be different between the RCTs and real-world practice [115,116]. Because the characteristics of patients in the RCTs do not necessarily match the standard population with T2DM, these results are difficult to be generalized. 

The Comparative Effectiveness of Cardiovascular Outcomes in New Users of SGLT2 inhibitors (CVD-REAL) Study is a multinational (USA, Sweden, Norway and Denmark) observational study. In this study, 309,056 patients newly initiated on either SGLT2 inhibitors, including empagliflozin, canagliflozin and dapagliflozin, or other glucose lowering drugs were analyzed on the risk for hospitalization for heart failure and death in patients with T2DM, using clinical data in real-world practice after propensity score matching [101]. There were 961 hospitalizations for heart failure cases during 190,164 person-years follow-up in 6 countries, and, of 215,622 patients in the United States, Norway, Denmark, Sweden and the United Kingdom, death occurred in 1334. The use of SGLT2 inhibitors was significantly associated with lower rates of hospitalization for heart failure and death, with no significant heterogeneity by country, compared to other glucose-lowering drugs (Table 3). In addition, the sub-analysis of the CVD-REAL study exhibited an association between the initiation of SGLT2 inhibitors versus other glucose-lowering drugs and the rates of MI and stroke. Overall, 205,160 patients were included, and, in the intent-to-treat analysis, over 188,551 and 188,678 person-years of follow-up (MI and stroke, respectively), there were 1077 MI and 968 stroke events. The initiation of SGLT2 inhibitors was associated with a modestly lower risk of MI and stroke [102] (Table 3).

The CVD-REAL2 study is a similar study design to the CVD-REAL and was conducted in six countries: South Korea, Japan, Singapore, Israel, Australia and Canada [117]. After propensity score matching, the risks for death, hospitalization for heart failure, MI and stroke were analyzed in 235,064 patients of each group, newly initiated on either SGLT2 inhibitors (dapagliflozin, empagliflozin, ipragliflozin, canagliflozin, tofogliflozin and luseogliflozin) or other glucose-lowering drugs. In total, 74% of patients had no history of CVD, and patient characteristics were well-balanced in both groups. The initiation of SGLT2 inhibitors significantly reduced the risk for death, hospitalization for heart failure, MI and stroke [117] (Table 3). 

In addition, a sub-analysis of the CVD-REAL-2 study showed that the initiation of SGLT2 inhibitors was associated with substantially lower risks of hospitalization for heart failure (HR 0.69, 95% CI 0.61–0.77; *p* < 0.0001), all-cause death (0.59, 0.52–0.67; *p* < 0.0001) and the composite of hospitalization for heart failure or all-cause death (0.64, 0.57–0.72; *p* < 0.0001) compared to dipeptidyl peptide-4 (DPP-4)-4 inhibitors [118]. Furthermore, in another retrospective cohort study, each of the 11,431 T2DM patients with peripheral artery disease (PAD) taking the SGLT2 inhibitor or DPP-4 inhibitor were analyzed for the risk for ischemic stroke and acute MI after propensity score matching. The use of the SGLT2 inhibitor had comparable risks of ischemic stroke and acute MI. However, the SGLT2 inhibitor group showed lower risks of congestive heart failure (HR: 0.66; 95% CI 0.49–0.89; *p* = 0.0062), lower limb ischemia requiring revascularization (HR: 0.73; 95% CI 0.54–0.98; *p* = 0.0367) or amputation (HR: 0.43; 95% CI 0.30–0.62; *p* < 0.0001) and cardiovascular death (HR: 0.67; 95% CI 0.49–0.90; *p* = 0.0089) compared to those in the DPP-4 inhibitor group. [119]. 

## 6. Perspectives and Conclusions 

In recent years, as effective anti-diabetic agents, SGLT2 inhibitors have not only achieved significant results in glycemic control via increasing urinary glucose excretion independent of insulin but have also presented cardiovascular protective effects, especially in reducing the risk of MACEs and hospitalization from heart failure. Although data from the CVOTs indicated less impressive results on atherosclerotic cardiovascular events such as MI and stroke than on heart failure, some real-world practice indicated the benefits on them (Table 2). This may be related to the significant heterogeneity in the CVD risk of the study populations [113]. Clinical studies have confirmed the cardiovascular safety of SGLT2 inhibitors [19,104,108,110]. Although some common adverse events, including polyuria, genital mycotic infections, urinary tract infections and ketoacidosis, need to be carefully monitored for [19,104,108], they will not cause severe or fatal consequences. Besides, although patients need to be alerted to some other risks, such as amputation and fractures related to canagliflozin [19], there have not been significantly higher incidences than of other anti-diabetic drugs, such as glucagon-like peptide 1 receptor agonists (GLP-1RA) and DPP-4 inhibitors [120]. Based on the safety, efficiency and cardiovascular benefits, pharmacologic recommendations from the guidelines of American Diabetes Association (ADA) recommend listing SGLT2 inhibitors for the treatment of T2DM patients with established high risk factors of atherosclerotic cardiovascular disease (ASCVD), ASCVD and heart failure [121]. The underlying mechanisms of SGLT2 inhibitors on cardioprotection may be related to improving the function of vascular endothelial cells, suppressing oxidative stress, inhibiting inflammation and regulating autophagy, which further protects from the progression of atherosclerosis. More studies are needed, however, to elucidate the underlying mechanisms.

## Figures and Tables

**Figure 1 jcm-11-00137-f001:**
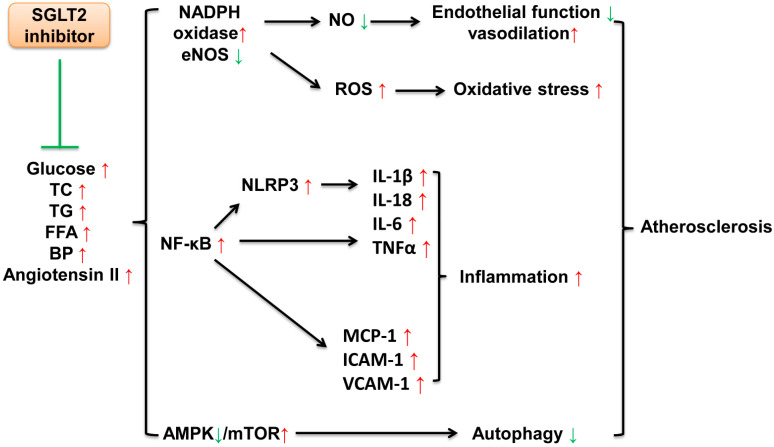
Mechanisms of SGLT2 inhibitors against atherosclerosis. Elevated metabolic characteristics including glucose, TC, TG, FFA, BP and angiotensin II accelerate the progression of atherosclerosis via the dysregulation of endothelial function, vasodilation and autophagy and activating oxidative stress and inflammation. SGLT2 inhibitors attenuate the alterations caused by these metabolic changes. TC, total cholesterol; TG, triglyceride; FFA, free fatty acid; BP, blood pressure; NADPH, nicotinamide adenine dinucleotide phosphate; NO, nitric oxide; eNOS, endothelial nitric oxide synthase; ROS, reactive oxygen species; NF-κB, nuclear factor kappa-B; NLRP3, nucleotide-binding oligomerization domain-like receptor family pyrin domain containing 3; IL, interleukin; TNFα, tumor necrosis factor α; MCP-1, monocyte chemoattractant protein 1; ICAM-1, intercellular cell adhesion molecule-1; VCAM-1, vascular cell adhesion protein 1; AMPK, AMP-activated protein kinase; mTOR, mechanistic target of rapamycin; ↑, increase; ↓, decrease.

**Figure 2 jcm-11-00137-f002:**
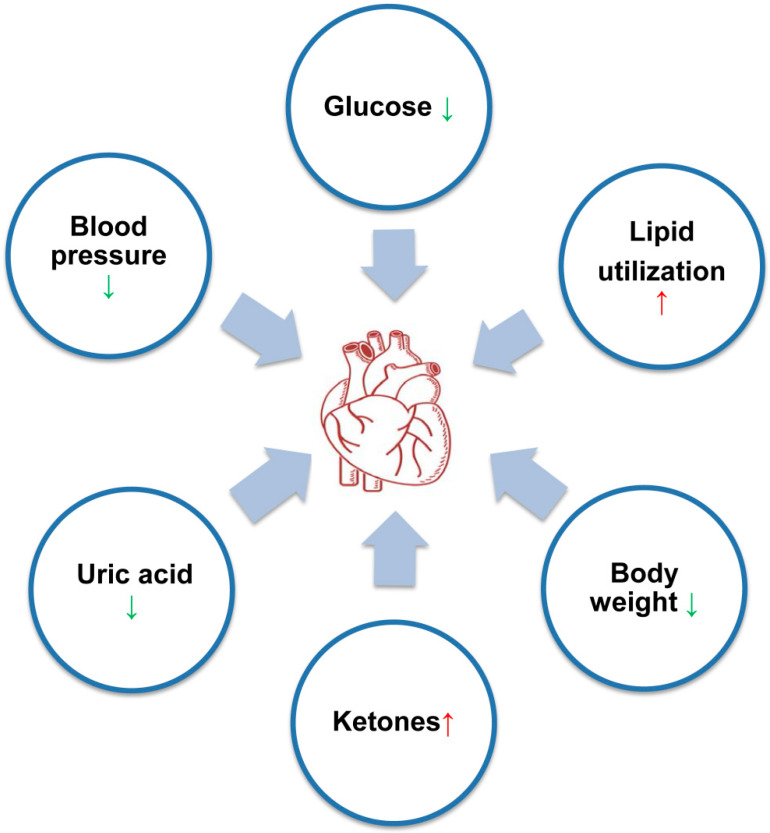
Improvements of some CVD-related clinical indicators by SGLT2 inhibitors. SGLT2 inhibitors not only reduce high glucose independent of insulin but also improve blood pressure, body weight, lipid profile and ketones and uric acid to present cardioprotective effects; ↑, increase; ↓, decrease.

**Table 1 jcm-11-00137-t001:** Benefits of SGLT2 inhibitors in atherosclerotic rodents.

SGLT2 Inhibitors	Atherosclerotic Rodents	Changes of Metabolic Characteristics	References
Dapagliflozin	STZ-induced diabetic ApoE^−/−^ mice	FBG↓, TC↓, TG↓ body weight↓, HbA1c↓	[34,73]
Dapagliflozin	STZ-induced diabetic Ldlr^−/−^ mice	FBG↓, TC↓, TG↓ body weight↓	[37]
Empagliflozin	HFD-fed ApoE^−/−^ mice	FBG↓, TC↓, heart rate↓, BP↓ TG↓, LDL↓, urinary microalbumin↓, body weight↓	[35,53,74]
Empagliflozin	STZ-induced diabetic ApoE^−/−^ mice	body weight↓, TG↓	[38]
Canagliflozin	HFD-fed diabetic ApoE^−/−^ mice	glucose↓, TC↓, TG↓, LDL↓, heart rate↓	[52]
Canagliflozin	STZ-induced diabetic ApoE^−/−^ mice	TC↓, glucose↓	[50]
Luseogliflozin	HFD-fed mice	body weight↓, TC↓, TG↓, BP↓, glucose↓	[75]
Luseogliflozin	STZ-induced diabetic ApoE^−/−^ mice	body weight↓, TC↓, TG↓, BP↓, glucose↓	[36]
Ipragliflozin	HFD-fed mice	body weight↓, glucose↓	[76]

↓, decrease; STZ, streptozotocin; Apo E, Apolipoprotein E; Ldlr, low density lipoprotein receptor; HFD, high fat diet; FBG, fasting blood glucose; TC, total cholesterol; TG, triglycerides; HbA1c, glycosylated hemoglobin; LDL, low-density lipoprotein; BP, blood pressure.

**Table 2 jcm-11-00137-t002:** Reported cardiovascular outcome trials of SGLT2 inhibitors.

	EMPA-REG OUTCOME [104]	EMPEROR- Reduced [105,106]	EMPEROR-Preserved [107]	CANVAS [19]	CREDENCE [14]	DECLARE- TIMI 58 [108]	DAPA-HF [109]	VERTIS CV [110]	SOLOIST-WHF [111]	SCORED [112]
Drug	empagliflozin	empagliflozin	empagliflozin	canagliflozin	canagliflozin	dapagliflozin	dapagliflozin	ertugliflozin	sotaglflozin	sotaglflozin
Patients	7020 (T2DM and CVD)	3730 (NYHA class II, III, or IV heart failure, EF ≤ 40%, 50% were DM)	5988 (NYHA class II-IV and EF ≥ 40%)	10,142 (T2DM)	4401 (T2DM and CKD)	17,160 (T2DM)	4744 (NYHA class II, III, or IV heart failure, EF ≤ 40%, 45% were T2DM)	8238 (T2DM and CVD)	1222 (T2DM and heart failure	10,584 (T2DM and CKD)
Duration of diabetes	≥10 years	-	-	13.5 ± 7.8	15.8 ± 8.6	11 (6–16)	-	12.9 ± 8.3	-	-
Median follow-up	3.1 years	16 months	26.2 months	126.1 weeks	2.62 years	4.2 years	3.5 years	3.5 years	9.0 months	16 months
Primary outcome *	0.86 (0.74–0.99)	0.76 (0.67–0.87)	0.79 (0.69–0.90)	0.86 (0.75–0.97)	0.70 (0.59–0.82)	0.93 (0.84–1.03)	0.74 (0.65–0.85)	0.97 (0.85–1.11)	0.67 (0.52–0.85)	0.74 (0.63–0.88)
Cardiovascular death	0.62 (0.49–0.77)	-	0.91 (0.76–1.09)	0.87 (0.72–1.06)	0.78 (0.61–1.00)	0.98 (0.82–1.17)	0.82 (0.69–0.98)	0.92 (0.77–1.11)	0.84 (0.58–1.22)	0.90 (0.73–1.12)
Nonfatal MI	0.87 (0.70–1.09)	-	-	0.85 (0.69–1.05)	-	0.89 (0.77–1.01)	-	1.04 (0.86–1. 27)	-	-
Nonfatal stroke	1.24 (0.92–1.67)	-	-	0.90 (0.71–1.15)	-	1.01 (0.84–1.21)	-	1.00 (0.76–1.32)	-	-
Death from any cause	0.68 (0.57–0.82)	-	1.00 (0.87–1.15)	0.87 (0.74–1.01)	0.83 (0.68–1.02)	0.93 (0.82–1.04)	0.83 (0.71–0.97)	0.93 (0.80–1.08)	0.82 (0.59–1.14)	0.99 (0.83–1.18)
Hospitalization for heart failure	0.65 (0.50–0.85)	0.70 (0.58–0.85)	0.71 (0.60–0.83)	0.67 (0.52–0.87)	0.61 (0.47–0.80)	0.73 (0.61–0.88)	0.70 (0.59–0.83)	0.70 (0.54–0.90)	0.64 (0.49–0.83)	0.67 (0.55–0.82)

EMPA-REG OUTCOME, Empagliflozin Cardiovascular Outcome Event Trial in Type 2 Diabetes Mellitus Patients–Removing Excess Glucose; EMPEROR-Reduced, Empagliflozin Outcome Trial in Patients with Chronic Heart Failure with Reduced Ejection Fraction; EMPEROR-Preserved, Empagliflozin Outcome Trial in Patients with Chronic Heart Failure with Preserved Ejection Fraction; CANVAS, Canagliflozin Cardiovascular Assessment Study; CREDENCE, Canagliflozin and Renal Events in Diabetes with Established Nephropathy Clinical Evaluation; DECLARE–TIMI 58, Dapagliflozin Effect on Cardiovascular Events–Thrombolysis in Myocardial Infarction 58; DAPA-HF, Dapagliflozin and Prevention of Adverse Outcomes in Heart Failure; VERTIS CV, Ertugliflozin Efficacy and Safety Cardiovascular Outcomes Trial; SOLOIST-WHF, the Effect of Sotagliflozin on Cardiovascular Events in Patients with Type 2 Diabetes Post Worsening Heart Failure; SCORED, the Effect of Sotagliflozin on Cardiovascular and Renal Events in Patients with Type 2 Diabetes and Moderate Renal Impairment Who Are at Cardiovascular Risk; T2DM, type 2 diabetes mellitus; CKD, chronic kidney diseases; NYHA, New York Heart Association; CVD, cardiovascular diseases; EF, ejection fraction; MI, myocardial infarction. * Primary outcomes of RCTs: EMPA-REG OUTCOME, CANVAS, CREDENCE, DECLARE–TIMI 58, VERTIS CV: MACE (the major adverse cardiovascular events including a composite of death from cardiovascular causes, nonfatal myocardial infarction or nonfatal stroke); EMPEROR-Reduced, EMPEROR-Preserved: the composite of cardiovascular death or hospitalization for heart failure; DAPA-HF: a composite of worsening heart failure (hospitalization or an urgent visit resulting in intravenous therapy for heart failure) or cardiovascular death; SOLOIST-WHF, SCORED: the total number of deaths from cardiovascular causes and hospitalizations and urgent visits for heart failure.

**Table 3 jcm-11-00137-t003:** Reported real world evidence of SGLT2 inhibitors for CVD.

	CVD-REAL [101]	Sub-Analysis of CVD-REAL [102]	CVD-REAL 2 [117]	Sub-Analysis of CVD-REAL 2 Study [118]	Retrospective Cohort Study on PAD Patients [119]
Drug	empagliflozin, canagliflozin and dapagliflozin	empagliflozin, canagliflozin and dapagliflozin	canagliflozin, dapagliflozin, empagliflozin, in all countries; ipragliflozin in South Korea and Japan; tofogliflozin, luseogliflozin in Japan	canagliflozin, dapagliflozin and empagliflozin in all countries apart from South Korea; ipragliflozin in South Korea and Japan; Tofogliflozin and luseogliflozin in Japan only	empagliflozin and dapa gliflozin
Patients	309,056	205,160	470,128	386,248	22,862
Regions	European and North American regions	European and North American regions	the Asia-Pacific, Middle East and North American regions	the Asia-Pacific, Middle East, European and North American regions	the Asia-Pacific region
Duration of diabetes	≥1 years	≥1 years	≥1 years	≥1 years	-
Mean follow-up *	239 days/ 211 days	254 days/ 232 days	374 days/ 392 days	Varied by country and regions	0.96 years/ 0.66 years
MI	-	0.85 (0.72–1.00)	0.81 (0.74–0.88)	0.88 (0.80–0.98)	0.84 (0.58–1.23)
Stroke	-	0.83 (0.71–0.97)	0.68 (0.55–0.84)	0.85 (0.77–0.93)	0.81 (0.62–1.06)
Death from any cause	0.49 (0.41–0.57)	-	0.51(0.37–0.70)	0.59 (0.52–0.67)	0.58 (0.49–0.67)
Hospitalizationfor heart failure	0.61 (0.51–0.73)	-	0.64 (0.50–0.82)	0.69 (0.61–0.77)	0.66 (0.49–0.89)

CVD-REAL, The Comparative Effectiveness of Cardiovascular Outcomes in New Users of SGLT2 inhibitors; PAD, peripheral artery disease; MI, myocardial infarction.* Mean follow-up: the mean follow-up time of SGLT2 inhibitors/ other glucose lowering drugs.

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
