# Peer review of "Effects of SGLT2 Inhibitors on Atherosclerosis: Lessons from Cardiovascular Clinical Outcomes in Type 2 Diabetic Patients and Basic Researches"

_jcm, 2021, doi:10.3390/jcm11010137_

Round 1
Reviewer 1 Report
In this manuscript, Xu et al reviewed the basic research and clinical data of SGLT2i in relation to the well-known benefits on atherosclerotic cardiovascular disease. Overall, it is an interesting paper, but several points should be addressed:
1. The English language should be reviewed. The manuscript contains many typo and grammar mistakes.
2. The structure of the review is a little bit strange. I would suggest begin with preclinical data (either in vitro and in vivo), followed by clinical mechanistic evidence, and finally conclude with CVOT and real-world data. I believe that in this manner, the reader would follow better the review.
3. It is surprising that, knowing the great effect of nephropathy in CVD risk, the information about SGLT2i on diabetic kidney disease is scarcely mentioned. I believe that this information is mandatory, at least a summary, in that review.
4. Why was sotagliflozin not included in the review? There are several hallmark studies that could have been included (SOLOIST-WHF trial, N Engl J Med 2021; 384:117-128; and SCORED trial; N Engl J Med 2021; 384:129-139).
5. In the same view, EMPEROR-Preserved is one of the hallmark studies that should also be included (Table 1)
6. Among the mechanisms thought to be responsible for the beneficial effects of SGLT2 inhibitors on the cardio-renal system, there is also a switch in the fuel used by cellules (from glucose to other more efficient, such as ketones, free fatty acids or branched-chain amino acids). The authors should also comment on this in the review.
Other comments:
1. Table 1: Since the primary outcome is different from one trial to another, it would be useful to include the specific primary outcome of each trial in the legend.
2. In lines 147-153, when the authors talk about ertugliflozin, seem to suggest that this drug was not approved by FDA and EMEA, but it has been approved years ago. Please, rewrite de paragraph to better reflect this fact.
3. Table 2: Why the other 2 references included in this section (43 and 44) were not summarized in the table? In the same table it would be welcomed that the authors included the countries of the patients included of each of the studies.
Author Response
Dear Editor of Journal of Clinical Medicine,
We thank reviewers and editors for proving us the chance to revise our manuscript. Authors are providing answers to all the comments by point to point raised by reviewers’ comments.
-----------------------------------------------------------------------------------------
Manuscript ID: jcm-1512851
Type of manuscript: Review
Title: Effects of SGLT2 inhibitors on atherosclerosis: lessons from cardiovascular clinical outcomes in type 2 diabetic patients and basic researches
Authors: Jing Xu, Taro Hirai, Daisuke Koya, Munehiro Kitada
Reviewer 1
In this manuscript, Xu et al reviewed the basic research and clinical data of SGLT2i in relation to the well-known benefits on atherosclerotic cardiovascular disease. Overall, it is an interesting paper, but several points should be addressed:
- The English language should be reviewed. The manuscript contains many typo and grammar mistakes.
----- We thank reviewer’s constructive comments on our manuscript and we are sorry for these mistakes on typo and grammar. We checked and corrected all wrong expressions in our manuscript.
- The structure of the review is a little bit strange. I would suggest begin with preclinical data (either in vitro and in vivo), followed by clinical mechanistic evidence, and finally conclude with CVOT and real-world data. I believe that in this manner, the reader would follow better the review.
----- We thank reviewer’s constructive comments on the structure. We rearranged the manuscript according to the reviewer’s comments, which begins with preclinical data, followed by clinical mechanistic evidence, and finally the CVOT and real-world data. Due to the change in the structure of the manucscript, the sequence of the tables and figures has also changed.
- It is surprising that, knowing the great effect of nephropathy in CVD risk, the information about SGLT2i on diabetic kidney disease is scarcely mentioned. I believe that this information is mandatory, at least a summary, in that review.
----- We thank reviewer’s constructive comments. Indeed, kidney disease is a key risk factor for CVD. A number of previous basic and clinical studies have also shown renoprotective effects of SGLT2i, especially in diabetic kidney disease. In this manuscript, we would like to mainly focus on the cardioprotective effects of SGLT2i. Therefore, we briefly summarized the role of SGLT2i in diabetic kidney disease in the Introduction (Line 46-60).
- Why was sotagliflozin not included in the review? There are several hallmark studies that could have been included (SOLOIST-WHF trial, N Engl J Med 2021; 384:117-128; and SCORED trial; N Engl J Med 2021; 384:129-139).
----- We thank reviewer’s constructive comments. We cited these two hallmark studies of sotagliflozin in Line 363-372 and Table 2.
- In the same view, EMPEROR-Preserved is one of the hallmark studies that should also be included (Table 1)
----- We thank reviewer’s constructive comments. We cited EMPEROR-Preserved trial in Line 332-334 and Table 2.
- Among the mechanisms thought to be responsible for the beneficial effects of SGLT2 inhibitors on the cardio-renal system, there is also a switch in the fuel used by cellules (from glucose to other more efficient, such as ketones, free fatty acids or branched-chain amino acids). The authors should also comment on this in the review.
----- We thank reviewer’s constructive comments. The effect of SGLT2i on the production of ketones may also be related to the potential mechanisms of cardiovascular protection. So we discussed this part in Line 237-248.
Other comments:
- Table 1: Since the primary outcome is different from one trial to another, it would be useful to include the specific primary outcome of each trial in the legend.
----- We thank reviewer’s constructive comments. Since the primary outcome of each study is not the same, we used “*” in the legend of Table 2 (Original Table 1) to explain the primary outcome of each study (Line 309-314).
- In lines 147-153, when the authors talk about ertugliflozin, seem to suggest that this drug was not approved by FDA and EMEA, but it has been approved years ago. Please, rewrite de paragraph to better reflect this fact.
----- We thank reviewer’s constructive comments. We modified the misunderstanding expression about ertugliflozin in this part (Line 356-358).
- Table 2: Why the other 2 references included in this section (43 and 44) were not summarized in the table? In the same table it would be welcomed that the authors included the countries of the patients included of each of the studies.
----- We thank reviewer’s constructive comments. We summarized references 43 and 44 (reference 119 and 120 after revision) in Table 3 (Original Table 2). Since some studies involve too many countries (more than 13 countries), we listed the regions enrolled by each study to succinctly describe the patients (Table 3 Regions).
Thank you for your time and consideration of our response to the critiques submitted by four reviewers. We will be happy to address any further questions that arise.
Sincerely,
Munehiro Kitada, M.D., Ph.D
kitta@kanazawa-med.ac.jp

Reviewer 2 Report
The review provides a nice overview of the current knowledge on the effect of SGLT2 inhibitors on cardiovascular events/adherosclerosis.
I have the following comments:
- Line 83: it should be European Medicines Agency (EMA) instead of European Union
- Line 92: there is a typing error in the word death
- Line 148: after Definition above, EMA can be used as an abrreviation
- Line 202 better rephrase heading for easier understanding, e.g.: Improvements of indicators related to CVD by SGLT2 inhibitors
- Line 229 should it Not be: benefit for patients with CVD?
- Figure 1: Part of the arrow next to Body weight is missing
- Figure 2: there is a typing error in “NADPH”. Perhaps the layout of this important figure could be made a little more attractive and easier to grasp.
- Among the cardiac effects of SGLT2 inhibitors, the effects on heart failure are most impressive; the effects on atherosclerosis (which are the main topic of the review), are also apparent in the larger studies but are less impressive than the effects on heart failure (as reported in Table 1). If the authors agree, this could be discussed more clearly, for example, in paragraphs 3. (Clinical evidence of SGLT2 inhibitors against CVD) and 6. (Perspectives and conclusion).
Author Response
Dear Editor of Journal of Clinical Medicine,
We thank reviewers and editors for proving us the chance to revise our manuscript. Authors are providing answers to all the comments by point to point raised by reviewers’ comments.
-----------------------------------------------------------------------------------------
Manuscript ID: jcm-1512851
Type of manuscript: Review
Title: Effects of SGLT2 inhibitors on atherosclerosis: lessons from cardiovascular clinical outcomes in type 2 diabetic patients and basic researches
Authors: Jing Xu, Taro Hirai, Daisuke Koya, Munehiro Kitada
Reviewer 2
The review provides a nice overview of the current knowledge on the effect of SGLT2 inhibitors on cardiovascular events/ atherosclerosis.
I have the following comments:
- Line 83: it should be European Medicines Agency (EMA) instead of European Union.
Line 92: there is a typing error in the word death.
Line 148: after Definition above, EMA can be used as an abbreviation.
Line 202 better rephrase heading for easier understanding, e.g.: Improvements of indicators related to CVD by SGLT2 inhibitors.
Line 229 should it Not be: benefit for patients with CVD?.
Figure 1: Part of the arrow next to Body weight is missing.
Figure 2: there is a typing error in “NADPH”. Perhaps the layout of this important figure could be made a little more attractive and easier to grasp.
----- We thank reviewer’s comments and we are sorry for the mistakes. We checked all the grammar and corrected the wrong expression in our manuscript (Line 94, the original Line 148 has been deleted following another reviewer’s comments, Line 212, Line 275, new Figure 1 and new Figure 2, respectively).
- Among the cardiac effects of SGLT2 inhibitors, the effects on heart failure are most impressive (as reported in Table 1). If the authors agree, this could be discussed more clearly, for example, in paragraphs 3. (Clinical evidence of SGLT2 inhibitors against CVD) and 6. (Perspectives and conclusion).
----- We thank reviewer’s constructive comments. We discussed the potential explanations in Line 373-379 and Line 434-437.
Thank you for your time and consideration of our response to the critiques submitted by four reviewers. We will be happy to address any further questions that arise.
Sincerely,
Munehiro Kitada, M.D., Ph.D
kitta@kanazawa-med.ac.jp

Round 2
Reviewer 1 Report
All my previous comments have been correctly addressed.